# STRUCTURE CONTROLLABLE TEXT GENERATION

## ABSTRACT

Controlling the presented forms (or structures) of generated text are as important as controlling the generated contents during neural text generation. It helps to reduce the uncertainty and improve the interpretability of generated text. However, the structures and contents are entangled together and realized simultaneously during text generation, which is challenging for the structure controlling. In this paper, we propose an efficient, straightforward generation framework to control the structure of generated text. A structure-aware transformer (**SAT**) is proposed to explicitly incorporate multiple types of multi-granularity structure information to guide the text generation with corresponding structure. The structure information is extracted from given sequence template by auxiliary model, and the type of structure for the given template can be learned, represented and imitated. Extensive experiments have been conducted on both Chinese lyrics corpus and English Penn Treebank dataset. Both automatic evaluation metrics and human judgement demonstrate the superior capability of our model in controlling the structure of generated text, and the quality ( like **Fluency** and **Meaningfulness**) of the generated text is even better than the state-of-the-arts model.

## 1 INTRODUCTION

Natural language is not just a sequence collections of tokens but a structure well-organized sequence expressing understandable information. The structure of language usually obeys a set of grammatical rules, which helps beginners grasp the language with less efforts. Similarly, incorporating the structure into neural language model can obtain an increasing abstract level of representation and improves the generalization which may potentially reduce the need of large amount of training data (Shen et al., 2019b). The incorporations of structure information demonstrates considerable improvements in many language understanding tasks (Zhang et al., 2019; Hao et al., 2019; Wang et al., 2019).

In text generation, it cares about not only the generated contents (i.e., what to say) but also the presented structure forms (i.e., how to say) (Peng et al., 2019). Similar contents or meanings can be presented with different structure forms. The structures and contents can be considered and planned separately to achieve a highly informative generated text. From an empirical view, controlling or planning the generated structure may be helpful in several aspects: i) reducing the uncertainty of the generated contents with specific structure conditions, which may contribute to a good quality of generated text; ii) enhancing the interpretability of the generated text since more controlling attributes can be realized during the generation; iii) improving the structure, format or style consistence in specific structure-constraint generation task or specific domain generation with particular formats, such as style or paraphrase generation (Chen et al., 2019; Ficler & Goldberg, 2017), poetry generation (Deng et al., 2020; Li et al., 2020), and lyric generation (Watanabe et al., 2018; Lu et al., 2019).

The language structures determined by the set of grammatical rules vary from different granularity levels, such as *participial construction* (*pc*) is character-level, *part of speech* (*pos*) is word/phrase level, and sequence length is sentence level. These kinds of structure are coupled and nested together, which are realized with the contents simultaneously in most of the token by token generation. It is difficult to disentangle the contents and the text structure, and even harder to discriminate and control the different granularity level of structure during text generation. Individually controlling some specific types of structure like *sequence length* (Kikuchi et al., 2016), *verbal predicate* (Tu et al., 2019) have been investigated in text generation. These works design specific structure representation and are inappropriately for controlling other types of structure, let alone controlling multiple

types of structure simultaneously. Directly embedding the structure and adding them into the word embeddings can achieve considerable controlling capability in character-level structure during text generation, such as tone level and rhyme (Deng et al., 2020) controlling in Chinese poetry generation. While this method may fail when the controlled structure (such as phrase level or sentence level) needs to aware the subsequent structure during the generation process. In addition to summarizing the structure embeddings and word embeddings, SongNet (Li et al., 2020) designs another structure embeddings which are queried and incorporated globally by the summarized embeddings to renew the representation. With pre-training and fine-tuning, the SongNet (Li et al., 2020) can also achieve good controllability in tailor-designed formats [1] (sentence level structure). The symbol sets for this format are particular designed and may not applicable for other type of structure.

Contrast to the above works, in this paper, we are not focus on controlling specific type of structure or format, instead we propose a framework to control more general types of structure in text generation. This framework allows for controlling individual type of structure, multiple or multi-granularity types of structure during text generation. The controlled types of structure are extracted from sequence templates (any valid sentence is a valid template) by one or several auxiliary models. The extracted structure information are regarded as conditions, and the auxiliary model can be any credible model or tool that can extract soundable structure information from template. Since we want the generation of the current token or word can aware the global structures, the bi-directional transformer encoder is adopted for structure representation and learning. The learned structure representations are further incorporated into the decoder to guide the realization of the controlled structure. The main contributions of this work are summarized as follows:

- A straightforward, interpretable structure controlling text generation framework is proposed, which is capable of controlling multi-granularity sequence structure from character-level to sentence-level structure by explicitly incorporating the corresponding structure information.

- A simple alignment method and structure embedding, representation and learning method are proposed, which are utilized for representing the multi-granularity and multiple types of structure.

- A structure-aware transformer language model is proposed, and the structure representation and token representation can be learned simultaneously. The structure information are queried globally and incorporated into the token representation with attention mechanism, which contribute to controlling the generated structure.

- Extensive experiments in controlling different individual type of structure and multi-granularity types of structure have been conducted on Chinese lyrics corpus. The structure controllability is effective and the quality of the generated lyrics is favorable. We also conduct controlling experiments on English Penn Treebank dataset, which demonstrates similar structure controlling capability with this proposed framework.

## 2 RELATED WORKS

Controllable text generation has received much attention recently. Many efforts are devoted to controlling the content of the generated text (Kiddon et al., 2016; Lebret et al., 2016; Shen et al., 2019a). Based on conditioned RNN language model, stylistic parameters are further incorporated as conditioning context to control stylistic aspects of the generated text (Ficler & Goldberg, 2017). Basing generator on VAEs, Hu et al. (2017) proposes a generative model to generate plausible sentences with designated semantics. A simple plug and play language model is proposed in Dathathri et al. (2019) to guide controlling attributes (e.g. topic or sentiment) in text generation, without further training of the pre-trained language model. None of these work attempts to control the structure of the generated text. A similar approach, exemplar-based text generation, is proposed in Peng et al. (2019), where for each input text, an exemplar text is retrieved from the training data and is then used to construct a customized decoder for outputting a target. It is ambiguously to discriminate how much the exemplar contributes to the generated structure or contents. Another similar work is SongNet (Li et al., 2020), which are proposed to control the so called rigid formats. The rigid for-

---

[1]This format or structure is more about the length of each sentence within one paragraph or passage.

mats are specifically designed with a sequence of placeholder symbols, which are utilized to control the sentence (or sub-sentence) length.

Our method is different from all the previous methods in fourfold: 1) we focus on a general structure controlling framework in text generation instead of controlling a specific type of structure;2) both individual type of structure and multiple or multi-granularity types of structure can be controlled;3) instead of designing the structure symbols by ourself, we adopt the most representative structure symbols as extracted by external models to increase the applicability of our framework;4) the extracted structure information decoupled from the sequence information are learned and represented fully before them are incorporated into word information to guide the text generation.

## 3 MODEL DESCRIPTION

### 3.1 STRUCTURE CONDITIONAL LANGUAGE MODEL

Given a natural language sequence denoted by $\boldsymbol{x} = [x_1, ..., x_T]$, each word denoted as $x_t, t = 1, ..., T$. The sequence joint distribution $p(\boldsymbol{x})$ can be factorized into the product of conditional distributions $p(x_t|x_{<t})$ as follows:

$$
\begin{aligned}
p(\boldsymbol{x}) =& p(x_1, ..., x_T) \\
=& \prod_{t=1}^{T} p(x_t|\boldsymbol{x}_{<t}).
\end{aligned}
\tag{1}
$$

A standard language model is modeling the above distribution and maximizing the corresponding likelihood accordingly (Bengio et al., 2003; Peters et al., 2018; Shen et al., 2019b). The above distribution considers the order structure of natural language sequence explicitly, and the conditional distribution are based on the previous word tokens.

Although the standard language model can generate sentence with high quality, the generated structure is inexplicable and cannot be controlled to satisfy specific generation task. Therefore, we incorporate the structure information explicitly into language model, and guide the structure generation. The joint distribution of sequence $\boldsymbol{x}$ can be reformulated as shown in Equation equation 2:

$$
\begin{aligned}
p(\boldsymbol{x}) =& p(x_1, ..., x_T) \\
=& p(\boldsymbol{s}) \prod_{t=1}^{T} p(x_t|\boldsymbol{x}_{<t}, \boldsymbol{s})
\end{aligned}
\tag{2}
$$

where, $\boldsymbol{s}$ represents the global structure of the natural language sequence $\boldsymbol{x}$, the global structure can be any of the structure information like pos tags or semantic roles of the sequence, and $p(\boldsymbol{s})$ is the prior distribution of the global structure. We extract the structure information with auxiliary model, and this structure information is considered as prior knowledge, which will not be optimized by the language model.

The model parameters are learned by maximizing the objective function of **SCLM**, which is to maximize the likelihood as shown in Equation equation 3:

$$
\max_{\theta} \log p_\theta(\boldsymbol{x}) = \sum_{t=1}^{T} \log p_\theta(x_t|\boldsymbol{x}_{<t}, \boldsymbol{s})
\tag{3}
$$

We utilize the *Transformer* (Vaswani et al., 2017) as the backbone for implementing our **SCLM**. The structure information is first extracted by auxiliary model and then encoded into transformer encoder. The structure information can be learned and represented fully, which can be further incorporated to contribute the aware of the structure for sequence token representation with attention mechanism. The reason why both the transformer encoder and decoder are adopted here is that we want each token in sequence to aware its local and global structure information. Only the *Transformer decoder*, like GPT (Radford et al., 2018) ignores the subsequent structure information of the token. The *Transformer* architecture is well designed and suitable for the implementation of the structure conditional language model. We only modified the input representation and few parameters of transformer.

## 3.2 STRUCTURE EXTRACTION

We use auxiliary model (such as lexical tool) $g(\bullet)$ to extract the structure information $\boldsymbol{s}$ from natural language sequence $\boldsymbol{x}$ as shown in Equation equation 4. The auxiliary model can be regarded as prior knowledge and will not be optimized.

$$\boldsymbol{s} = g(\boldsymbol{x}). \tag{4}$$

The structure can be any sounded structure information of language sequence vary from character-level structure (like *participial construction*), word-level structure (like *part of speech*) to sentence-level structure (*positions* for example).

The multi-granularity types of sequence structure $\boldsymbol{s_1}, \boldsymbol{s_2}, .., \boldsymbol{s_i}$ can be extracted by different auxiliary models $g_1(\bullet), g_2(\bullet), ..., g_i(\bullet)$ respectively. Since each structure unit (especially for word-level and sentence-level structure) may contain several characters, we assign these characters with the same symbol of this kind structure. We keep the length of the structure the same with the sequence tokens.

To be specific, we use the *part of speech* (*pos*) and *participial construction* (*pc*) as examples to illustrate the alignment of multi-granularity types of structure. The *pos* information can be extracted by many lexical analyzer tools like Jieba analyzer and Stanza (Qi et al., 2020) for Chinese and English sequence respectively. In Chinese, the *pos* is a type of word-level structure, and the *participial construction* is the character-level structure for each segmented word. We utilize the symbol collections $\mathbb{C}_{pos} = \{n, v, r, ...\}$ [2] from lexical analyzer (like Jieba) to represent the *pos* for each word. The symbol collections $\mathbb{C}_{pc} = \{P, S, B, M, E\}$ [3] are utilized to represent the *pc* for each character within each word. Suppose we have two levels (word-level and character-level) structure information for a sequence $\boldsymbol{x} = [x_1, ..., x_i, ..., x_n]$, we can also present the word-level form of the sequence with $\boldsymbol{w} = [w_1, ..., w_j, ..., w_{n_w}], n_w \leq n$, and the *pos* structure can be represented with $\boldsymbol{s'_w} = [pos_1, ..., pos_j, ..., pos_{n_w}], pos_j \in \mathbb{C}_{pos}$; each word contains several characters $w_j = [..., x_{j,k}, ...], k \in [1, m_j]$, and the *pc* structure for each word are $\boldsymbol{s}_{c,j} = [..., pc_{j,k}, ...], pc_{j,k} \in \mathbb{C}_{pc}$ where $\sum_{j=1}^{n_w} m_j = n$. Therefore, we can obtain the word-level structure (*pos*) and character-level structure (*pc*) with the same length with the original sequence as can be shown in the following expressions:

$$\boldsymbol{s}_w = [..., \underbrace{pos_j, .., pos_j}_{m_j}, ...], j \in [1, n_w] \tag{5}$$

$$\boldsymbol{s}_c = [..., \underbrace{pc_{j,1}, ..., pc_{j,k}, ..., pc_{j,m_j}}_{m_j}, ...] \tag{6}$$

The sentence level structure like positions have unique representation for each token and do not need any further processing for the alignment. With the alignment process, multi-granularity and multi-type of sequence structure can be incorporated and controlled in the generation.

An illustration of multi-granularity structure information for a natural language sentence can be shown in Fig. 1.

## 3.3 STRUCTURE AWARE TRANSFORMER

We propose a **Structure Aware Transformer** (**SAT**) to implement the multi-granularity structure controlling in text generation. The encoder stacks multi-layer *Transformer encoder* (Vaswani et al., 2017) with **Multi-Head Self Attention** in each layer to represent the extracted structure. The extracted structure information are first embedded and then summarized together as the structure input representation $\boldsymbol{H_0}$ [4], which allows for controlling multiple types of structure in text generation simultaneously. The structure representation for each layer $\boldsymbol{H_{l_e}}, l_e = 1, ..., N_e$ can be obtained

---

[2] $n, v, r$ represent the noun, verb, pronoun respectively; for complete symbols can refer to https://github.com/fxsjy/jieba.

[3] $P$ represent the *pc* structure of special token, $S$ represent a word only contains a single character, $B, M, E$ represent the beginning, middle and ending of the word respectively.

[4] positions are regarded as sentence-level structure and are also added into the structure representation.

|          | I / love / chips 我 / 喜欢 / 炸薯条 |       |         |
|----------|:--------:|:-----:|:-------:|
| **Sequence** | 我 / 喜欢 / 炸薯条 |       |         |
| **pc**   | S        | B E   | B M E   |
| **pos**  | r        | v     | n       |

Figure 1: An alignment illustration of word-level and character-level structure for a Chinese sequence. The final *pc* structure is $\boldsymbol{s}_{pc} = [S, B, E, B, M, E]$, and the final *pos* structure is $\boldsymbol{s}_{pos} = [r, v, v, n, n, n]$.

according to the following formulas:

$$\boldsymbol{H}_0 = \sum_{i=0}^{m} \boldsymbol{E}_{\boldsymbol{s}_i}(\boldsymbol{s_i}) \tag{7}$$

$$\begin{pmatrix} \boldsymbol{Q_s} \\ \boldsymbol{K_s} \\ \boldsymbol{V_s} \end{pmatrix} = \boldsymbol{H}_{l_e-1} \begin{pmatrix} \boldsymbol{W_s^q} \\ \boldsymbol{W_s^k} \\ \boldsymbol{W_s^v} \end{pmatrix} \tag{8}$$

$$\boldsymbol{A_s} = softmax(\frac{\boldsymbol{Q_s}\boldsymbol{K_s}^T}{\sqrt{d}})\boldsymbol{V_s} \tag{9}$$

$$\boldsymbol{H}'_{l_e} = LN(\boldsymbol{A_s} + \boldsymbol{H}_{l_e-1}) \tag{10}$$

$$\boldsymbol{H}_{l_e} = LN(FFN(\boldsymbol{H}'_{l_e}) + \boldsymbol{H}'_{l_e}) \tag{11}$$

where $\boldsymbol{E}_s$ is the structure embedding matrix, $m$ is the number of structure types, $l_e$ is the number of encoder layers, and $\boldsymbol{H}_{l_e}$ is the output structure representation for layer $l_e$. *softmax*($\bullet$), *LN*($\bullet$), *FFN*($\bullet$) represent the softmax function, layer normalization and feed-forward network respectively.

The final layer output of structure encoder $\boldsymbol{H}_{N_e}$ is then utilized by the decoder, and the decoder is similar to the *Transformer decoder* (Vaswani et al., 2017) with two attention blocks in each layer. The below attention block is a **Masked Multi-Head Self Attention**, which obtains the token $x_t$ representation without considering the information from its subsequent tokens $\boldsymbol{x}_{>t}$. The upper attention block is the **Structure-Aware Attention**, which incorporates the structure information ($\boldsymbol{H}_{N_e}$) into the token representation.

$$\boldsymbol{F}_0 = \boldsymbol{E}_x(\boldsymbol{x}) + \boldsymbol{E}_p \tag{12}$$

$$\boldsymbol{F}'_{l_d} = \boldsymbol{Mask\text{-}Att}(\boldsymbol{F}_{l_d-1}) \tag{13}$$

$$\boldsymbol{Q} = \boldsymbol{F}'_{l_d}\boldsymbol{W}^q \tag{14}$$

$$\boldsymbol{K}, \boldsymbol{V} = \boldsymbol{H}_{N_e}\boldsymbol{W}^k, \boldsymbol{H}_{N_e}\boldsymbol{W}^v \tag{15}$$

$$\boldsymbol{A}_{sx} = softmax(\frac{\boldsymbol{Q}\boldsymbol{K}^T}{\sqrt{d}})\boldsymbol{V} \tag{16}$$

$$\boldsymbol{F}''_{l_d} = LN(\boldsymbol{A}_{sx} + \boldsymbol{F}'_{l_d}) \tag{17}$$

$$\boldsymbol{F}_{l_d} = LN(FFN(\boldsymbol{F}''_{l_d}) + \boldsymbol{F}''_{l_d}) \tag{18}$$

where $\boldsymbol{E}_x$ is the token embedding matrix, $\boldsymbol{E}_p$ is the position embedding matrix, *Mask-Att* represents the **Masked Multi-Head Self Attention** mechanism, $l_d \in [1, N_d]$ is the number of decoder layer, $\boldsymbol{F}_{l_d}$ is regarded as the structure-aware token representation.

The final output of the decoder $\boldsymbol{F}_{N_d}$ can be utilized to calculate the probabilities $p_\theta(x_t|\boldsymbol{x}_{<t}, \boldsymbol{s})$, and the parameters in the architecture can be learned by maximizing the likelihood in Equation equation 3.

### 3.4 STRUCTURE CONTROLLABLE GENERATION

With our proposed **SAT**, we can controlling multi-granularity and multiple types of structure in text generation simultaneously. Both the specified structure information $\boldsymbol{s}$ and the template sequence $\boldsymbol{x}$

can be utilized to control the structure of the generated text. The *input context* $x_c$, which can be any precede words for continue generation (or topic words for topic related text generation), is utilized to guide the content generation. If no *input context* is specified, the model will start the generation from start token until the end token is generated.

## 4 EXPERIMENTS AND EVALUATIONS

### 4.1 SETUP

We follow the GPT2 source code from *huggingface repository* (Wolf et al., 2019) and add an additional structure encoder with multi-head self attention to implement our proposed **SAT**. The number of encoder layer $N_e$ is 2,[5] and the number of decoder layer $N_d$ is 6. The other configurations are the same with the GPT2 except vocabulary size and structure size for embedding matrix. The structure information is extracted by Jieba [6] and Stanza (Qi et al., 2020) for Chinese and English text respectively. The extracted structure information is reagrded as the conditional structure, which are not optimized by our language model. However, the structure embeddings ($E_s$) or representation vector ($H_{l_e}$) can be learned by the proposed **SCLM**.

### 4.2 DATASETS

We conduct the experiments on both Chinese lyrics corpus and English Penn Treebank (**PTB**) dataset. Over 80,000 Chinese lyrics are crawled from a set of online music websites, and the number of lyrics sentences without repetition is about 1.38 million. Every two adjacent lyric lines within one song are concatenated with comma to increase the structure complexity, which is prepared for the generation task. We randomly split them into three parts for model training(90%), validation(5%) and testing(5%). The statistics of data corpus for Chinese Lyrics and **PTB** dataset are shown in Appendix.

### 4.3 MODEL COMPARISONS

We conduct the model comparisons on both Chinese lyrics corpus and English **PTB** dataset. The *pos* structure is considered as the mainly structure for the structure conditional language model, and we compare the **GPT2** and **SAT-*pos*** on the continue text generation with both Chinese lyrics corpus and **PTB** dataset. The continue text generation utilizes the *prompt* words to guide the following sequence generation. The length of each *prompt* is randomly varied from 0 [7] to the half length of the whole template sequence.

We also investigate multi-granularity types of structure individually and simultaneously for the **S-CLM** on Chinese lyrics corpus. The additional structure is the *participial construction* (*pc*), which can also be extracted by Jieba analyzer. Two other models **SAT-*pc*** (conditioned with *pc* structure) and **SAT-$p^2$** (conditioned with both *pc* and *pos* structure) are also compared on Chinese lyrics corpus. To better compare the generation capability of these language models, a topic related generation task are also performed based on Chinese lyrics corpus. The topic words are extracted by Jieba with **TF-IDF** method. For fair comparisons, we train the **SAT** and **GPT2** from scratch without utilizing any pre-trained model.

### 4.4 EVALUATION METRICS

Both automatic evaluation metrics and human evaluations are adopted for model comparisons. The **PPL** is to evaluate the performance of language model, and the **BLEU** score (Papineni et al., 2002) is utilized to measure the content similarity of the generated text with its referred sequence text.

The structure controlling capability, like the sentence length, the *pos* and participial construction are also compared. The length controllability is measured by the prediction accuracy. Assume the length

---

[5] We have conduct experiments on different number of encoder layers, and the gain of larger number of layer is trivial, please refer to Appendix for the result and analysis.

[6] https://github.com/fxsjy/jieba

[7] Indicators no prompt word is specified, and the generation starts from the start token.

of the input template is $l$ and the predicted sequence length is $l'$. If the length difference $\delta = |l - l'|$ is within specified threshold, we regard the predicted length is accurate with this tolerance. We report the **Accuracy** of length control with tolerance $\delta \leq 0$, $\delta \leq 2$ and $\delta \leq 4$.

The **BLEU** score can also be utilized to measure the *pos* and *pc* controllability. We extract the *pos* and *pc* structure from both test template and predicted sequence with the same lexical tool (Jieba or Stanza), and the **BLEU** score of *pos* or *pc* can be calculated accordingly.

Human evaluation is inevitable for evaluating the quality of the generated text, especially in the meaningfulness and fluency. However, human evaluation is time-consuming and costing. We conduct the human evaluation for model comparisons on the continue generation task of Chinese lyrics. Four well educated annotators are recruited to evaluate the continue generation of Chinese lyrics sentence in three dimensions, namely **Fluency**, **Meaningfulness** and **Structure Matching**. The **Fluency** and **Meaningfulness** are easy to understand and have been utilized by many previous works Deng et al. (2020). The **Structure Matching** is to evaluate the matching degree of generated text structure and template structure in several aspects, which considers the global structure (like subjective, predicates and objective structure) matching , constitute structure matching and *pos* matching for local words. The rating scores are 1 to 5 to represent the quality from bad to excellent for all the criteria. Each model generates 500 lyric lines and with the same random length *prompt*. Total 1000 lyric lines are generated and randomly shuffled, and the four annotators rated on the shuffled lyrics lines. Therefore, we can obtain 4000 ($4 \times 2 \times 500$) ratings.

## 4.5 RESULTS & DISCUSSIONS

Table 1 shows the perplexity of the language models on both Chinese lyrics corpus and English **PTB** dataset. The results demonstrates that the *pos* structure can improve the language modeling performance on both Chinese and English sequence. And the language model performance can be further improved when additional structures are also incorporated, as shown in the table that the **PPL** of **SAT**-$p^2$ with the lowest scores. We can observe that the *pos* condition gains more improvements than the *pc* structure condition when compared **SAT**-*pos* model with **SAT**-*pc* model on Chinese lyrics corpus. The probably reason is that the *pos* structure (with dictionary size 58) contains richer structure information than *pc* structure (with dictionary size 5).

Table 1: *Perplexity* scores for model comparisons.

| Model | Chinese Lyrics | | PTB | |
|---|---|---|---|---|
| | Val. | Test | Val. | Test |
| GPT2 | 10.57 | 11.24 | 8.60 | 8.12 |
| SAT-*pc* | 7.51 | 8.01 | – | – |
| SAT-*pos* | 4.07 | 4.34 | **3.56** | **3.39** |
| SAT-$p^2$ | **3.92** | **4.19** | – | – |

The text generation performance can also be improved by our proposed model, as demonstrated in Table 5, and 2. The generation performance of our proposed structure conditional models obtains obvious improvements on the **BLEU** scores of text sequence. The improvements of text **BLEU** scores are with similar improvement paradigms as the **PPL** scores, which are 1) the prior structure information is useful for the modeling and generation; 2) the more the structure information incorporated, the better the modeling performance and generation results. Our proposed model **SAT** shows the superior structure controllability as demonstrated by the **BLEU** scores on *pc* and *pos* structure. The **BLEU** scores of structure can be significantly improved when the corresponding structures are conditioned and incorporated into the language model.

It is interesting to observe that the *pos* structure can improve the **BLEU** scores on *pc* significantly (**SAT**-*pos* versus **GPT2**), while the *pc* structure only slightly improves the **BLEU** scores on *pos* (**SAT**-*pc* versus **GPT2**). These phenomena are consistent with the fact that the *pos* structure can reflect the segmentation border of words. The *pc* structure is more coarse structure information than *pos*. We also observe that the *pos* structure can not improve the **BLEU** scores on *pc* when the *pc* structure is already incorporated (**SAT**-$p^2$ versus **SAT**-*pc*), while the *pc* structure can further improve **BLEU** scores when the *pos* structure is already incorporated (**SAT**-$p^2$ versus **SAT**-*pos*).

Table 2: The **BLEU** scores for model comparisons on the continue generation of Chinese lyrics.

| Task | Model | Text | | *pc* | | *pos* | |
|---|---|---|---|---|---|---|---|
| | | **BL-1** | **BL-2** | **BL-1** | **BL-2** | **BL-1** | **BL-2** |
| **Continue** | **GPT2** | 0.144 | 0.015 | 0.653 | 0.608 | 0.396 | 0.241 |
| | **SAT-*pc*** | 0.174 | 0.028 | **0.98** | **0.975** | 0.486 | 0.323 |
| | **SAT-*pos*** | 0.268 | 0.115 | 0.939 | 0.919 | 0.949 | 0.939 |
| | **SAT-$p^2$** | **0.269** | **0.115** | 0.97 | 0.962 | **0.952** | **0.941** |
| **Topic** | **GPT2** | 0.247 | 0.133 | 0.676 | 0.643 | 0.449 | 0.309 |
| | **SAT-*pc*** | 0.279 | 0.151 | **0.981** | **0.976** | 0.553 | 0.395 |
| | **SAT-*pos*** | 0.356 | 0.226 | 0.936 | 0.915 | 0.941 | 0.925 |
| | **SAT-$p^2$** | **0.36** | **0.231** | 0.966 | 0.957 | **0.948** | **0.935** |

The probably explanation is that the *pos* structure is a type of fine-grained (or micro scale) structure compares to the *pc* structure, and the fine-grained information is too details for clarifying coarse information [8].

The length controllability of our proposed model is demonstrated by Table 6 (in Appendix). Although the text length is not explicitly incorporated as the condition, the generated text length is controlled effectively by the sequence length of conditioned *pos* and *pc*.

The Human evaluation results, as shown in Table 3, also demonstrate that the proposed model is superior in controlling the structure of the generated text. Although the strict structure constraints, our model can also achieve even better performance in terms of *Fluency* and *Meaningfulness*. As for the case and ablation studies please refer to the Appendix.

Table 3: The Human evaluation results for continue generation of Chinese lyrics. **Flu.**, **Mea.**, **Mat.** represent the *Fluency*, *Meaningfulness* and *Structure Matching*.

| Model | Flu. | Mea. | Mat. |
|---|---|---|---|
| **GPT2** | 3.12 | 3.25 | 2.01 |
| **SAT-$p^2$** | **3.59** | **3.82** | **4.01** |

## 5 CONCLUSION

In this paper, we propose a straightforward, interpretability and effective framework to control a wide range of language structures from character-level, word-level to sentence-level structure in text generation. These kinds of structure regarded as prior knowledge are explicitly extracted by external models and aligned together, which allows for both individually and simultaneously controlled in text generation. The structures are decoupled from word information and the structure representations are learned by bi-directional transformer encoder, which is powerful to learn the structure representations sufficiently. Subsequently, the structure representations are globally queried by the transformer decoder and are incorporated into contextualized word representations to guide the text generation with corresponding types of structure.

Extensive experiments on both Chinese lyrics corpus and English Penn Treebank dataset have been conducted. Without pre-training on large amount of dataset, the results demonstrate the powerful structure controllability of our method in terms of the sequence length, *pos*, and *pc*. The superior performance of text quality with respect to *fluency* and *meaningfulness* are also achieved significant improvements than the free text generation model. Our method can be easily applied to control other kinds of structure in text generation and may even reduce the uncertainty and improve the quality of the generated text.

---

[8]Let's analogy this with an example, the micro-scale shape of an object is not helpful or may even disturbing the identification of a macro-scale shape of the object.

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

## A  DATA DESCRIPTION

The statistic of utilized data is summarized in Table 4. The size of *pos* structure for Chinese lyrics datasets extracted from Jieba is 58, and the size of *participial construction* for lyrics words is 5. The vocabulary size of the lyrics dataset including the special tokens (like *[PAD]*, *[START]*, *[END]*, *[TOPIC]*) is 4102, and some low frequency characters are replaced with *[UNK]*. The special tokens that indicate the start, end or pad of the sentence are regarded as a special structure. The vocabulary size of **PTB** is 10005 (with some special tokens), and the structure size of *pos* extracted by Stanza is 43.

Table 4: The Statistics of the utilized data corpus.

| Corpus | #Train | #Validation | #Test |
|---|---|---|---|
| Chinese Lyrics | 6088,90 | 33830 | 33902 |
| Penn Treebank | 42068 | 3370 | 3761 |

## B  SUPPLEMENTARY RESULTS

Table 5: The **BLEU** scores for model comparisons on the continue generation of **PTB** dataset.

| Model | Text | | *pos* | |
|---|---|---|---|---|
| | BL-1 | BL-2 | BL-1 | BL-2 |
| GPT2 | 0.093 | 0.028 | 0.277 | 0.128 |
| SAT-*pos* | **0.309** | **0.138** | **0.903** | **0.788** |

Table 6: The length accuracy results on both Chinese lyrics test corpus and PTB test datasets. **Topic** in the bracket represent the topic related generation task, and **Continue** in the bracket represents the continue generation task.

| Model | Lyrics (Topic) | | | Lyrics (Continue) | | | PTB (Continue) | | |
|---|---|---|---|---|---|---|---|---|---|
| | $\delta \leq 0$ | $\delta \leq 2$ | $\delta \leq 4$ | $\delta \leq 0$ | $\delta \leq 2$ | $\delta \leq 4$ | $\delta \leq 0$ | $\delta \leq 2$ | $\delta \leq 4$ |
| GPT2 | 0.08 | 0.37 | 0.61 | 0.10 | 0.47 | 0.73 | 0.11 | 0.36 | 0.56 |
| SAT-*pc* | 1.00 | 1.00 | 1.00 | 1.00 | 1.00 | 1.00 | – | – | – |
| SAT-*pos* | 1.00 | 1.00 | 1.00 | 1.00 | 1.00 | 1.00 | 0.91 | 1.00 | 1.00 |
| SAT-$p^2$ | 1.00 | 1.00 | 1.00 | 1.00 | 1.00 | 1.00 | – | – | – |

## C CASE STUDY

Figure 2 and 3 compare several cases generated by **GPT2** and our proposed **SAT** on Chinese lyrics corpus and **PTB** dataset. We can notice that our model is capable of controlling the sentence-level structure (*length*), word-level structure (*pos*) (and character-level structure *pc* for lyrics generation) simultaneously, and the quality of the generated texts are also qualified and understandable. It is should be noted that the auxiliary model or tool utilized for extracting the structure is not optimized by our language model, and the accuracy of these tool will affect the quality of the generated text.

| | Prompt | Prediction | pc | pos |
|---|---|---|---|---|
| **Template :** | 蓦然回首 情 / 已远 /, / 身不由已 / 在 / 天边
Suddenly looking back the love is far away,
and I can't help myself. | B M M E / S / B E / S / B M M E / S / B E | ns / n / d / x / i / p / s |
| **SAT-*pc* :** | 蓦然回首 的 / 情歌 /, / 千言万语 / 诉 / 衷肠
A love song for looking back,
thousands of words are spoken sincerely. | B M M E / S / B E / S / B M M E / S / B E | ns / uj / n / x / i / vn / n |
| **SAT-*pos* :** | 蓦然回首 情 / 不断 /, / 千言万语 / 在 / 心头
Suddenly looking back the infinite love,
thousands of words in my heart. | B M M E / S / B E / S / B M M E / S / B E | ns / n / d / x / i / p / s |
| **SAT-$p^2$ :** | 蓦然回首 泪 / 不停 /, / 千言万语 / 在 / 心间
Suddenly looking back in tears,
thousands of words in my heart. | B M M E / S / B E / S / B M M E / S / B E | ns / n / d / x / i / p / s |
| **GPT :** | 蓦然回首 望 / 着 / 你 / 的 / 脸 /, / 在我心中 / 永远 / 不 / 分离
Suddenly looking back at your face,
you will never be separated in my heart. | B M M E / S / S / S / S / S / B M M E / B E / S / B E | ns / v / uz / r / uj / n / x / i / d / d / v |

Figure 2: Cases generated by different models with the same input *prompt*.

| | Prompt | Prediction |
|---|---|---|
| **Template :** | | a dog was running in a room
(DT NN VBD VBG IN DT NN) |
| **SAT-*pos* :** | | the market was falling at this point
(DT NN VBD VBG IN DT NN) |
| **GPT :** | | the company said the new facility
will begin to earnings for fiscal N |
| **Template :** | a
DT | dog is walking in a room
(NN VBZ VBG IN DT NN) |
| **SAT-*pos* :** | a
DT | problem is coming at that rate
(NN VBZ VBG IN DT NN) |
| **GPT :** | a | unk spokesman said the company
is still trying to sell N million australian
dollars us $ N billion of assets |
| **Template :** | the cat
(DT NN | is walking in the room
VBZ VBG IN DT NN) |
| **SAT-*pos* :** | the woman
(DT NN | is working on the problem
VBZ VBG IN DT NN) |
| **GPT :** | the woman | says mr bush is n't worried whether
a unk in this way about taking |

Figure 3: Cases comparisons on different models with different length of *prompt* and different templates. The template is utilized to provide *pos* information for **SAT-*pos***, and the corresponding *pos* are in the bracket. The notations of the *pos* are provided by Stanza.

# D    ABLATION STUDY

We conduct ablation study experiments on Chinese lyrics corpus to investigate the effects of encoder layer number. The **PPL** scores are compared for layer 2, 4, and 6 for different types of structure information. As shown in table 7, we cannot observe the obvious improvement due to the larger number of encoder layer. The probably explanation is that the structure information is comparative small (with dictionary size 5 for *pc* structure and 58 for *pos*, while vocabulary size is 4102) and 2 layer encoder is enough to process nd represent the information.

Table 7: *Perplexity* scores of different encoder layer number for different structure information.

| Model | 2 layer | | 4 layer | | 6 layer | |
|---|---|---|---|---|---|---|
| | Val. | Test | Val. | Test | Val. | Test |
| **SAT-*pc*** | 7.508 | **8.013** | **7.491** | 8.606 | 8.105 | 8.686 |
| **SAT-*pos*** | 4.068 | **4.342** | **4.058** | 4.765 | 4.554 | 4.847 |
| **SAT-$p^2$** | 3.923 | **4.190** | **3.912** | 4.563 | 4.310 | 4.604 |

