# OpenReview forum: "Structure Controllable Text Generation"
_ICLR.cc/2021/Conference — Reject_

### Official Review · AnonReviewer4 · 2020-10-27
**Official Blind Review #4**

**Rating:** 3
**Confidence:** 4

**Review:**

Review of Paper: Structure Controllable Text Generation

Summary:
This paper proposes a structure-aware Transformer (SAT) by incorporating multiple types of multi-granularity structure information to control the text generation. Their method can extract structure information given sequence template by auxiliary model and learn the structure representations.

Strengths:
•	The proposed model is able to control multiple or multi-granularity types sequence structure from character-level to sentence-level structure by explicitly incorporating the corresponding structure information.
•	The model can simultaneously learn the structure representation and token representation.

Weakness:
•	The paper does not clearly define the input content of this task.
What’s the input of the task? It seems the input should be the “prompt” (prefix of the poem), according to figure 2 in appendix C.

If so, why not use the content of “prompt” to predict the following tokens, which is more important for generation rather than the structure information?

If the input of this task is the structure information of the whole poem, the model can access future information (the structure of the sequence to be generated) during inference. It is unfair to compare this model with GPT-2, because it’s not a standard language modeling task. Instead, it should compare with structure controlled text generation task like SongNet (Li, P.), but the paper does not conduct such a comparison.

•	The baseline is not solid and suitable for the datasets they use.
GPT-2 is suitable for large-scale training corpus. Training GPT-2 on a small-scale dataset may result in lower performance, so GPT-2 cannot act as a strong baseline on a small-scale dataset (e.g. Chinese lyrics corpus). However, GPT-2 is the only one baseline in this paper.

As a large-scale text generation dataset is not hard to acquire, why not try it on a large-scale dataset?





Reference:
[Li, P.] Li, P., Zhang, H., Liu, X., & Shi, S. (2020). Rigid Formats Controlled Text Generation. arXiv preprint arXiv:2004.08022.

---

### Official Review · AnonReviewer2 · 2020-10-27
**Official Blind Review #2**

**Rating:** 2
**Confidence:** 5

**Review:**

SUMMARY

This paper presents a text generation model conditioned on desired structures. The proposed method is essentially a translation model from structure information (represented with multiple sequences of tokens) to a text. This study converts a text into structure information such as part of speech (POS) and participial construction (PC). Then, this paper proposes Structure Aware Transformer (SAT), which is essentially the same as the Transformer architecture. The experiments use datasets of Chinese lyrics and English Penn Treebank. This paper reports that giving structure information improved the performance in PPL and BLEU compared with GPT-2.

PROS

It was nice to confirm that we can control language generation from POS and PC.

CONS

The proposed method presented in Section 3.3 is identical to Transformer except that:

+ An input consists of multiple sequences of structure information (e.g., pos and pc)

+ Input embeddings are sums of structure embeddings (Equation 7)

For this reason, I do not think Structure Aware Transformer (SAT) is a novel proposal. If this explanation is sufficient, I think that the descriptions in Section 3.1 and 3.3 are redundant.

Because structure sequences (POS and PC) are obtained from sentences in the test set, it is not surprising to see performance improvements of language models. In other words, predicting word sequences with some hints (POS and PC) is much easier than doing without any hint (GPT-2). For this reason, the findings in this paper are not convincing.

QUESTIONS

Could you explain the major difference between the proposed method and Transformer (excluding minor differences in how input embedings are computed and hyper-parameters such as the number of layers)?

---

### Official Review · AnonReviewer1 · 2020-10-28
**Recommendation to Reject**

**Rating:** 2
**Confidence:** 5

**Review:**

Summary:
The paper proposes a variation of the conditioned Transformer-based language model. The authors use POS labels (for English and Chinese) and participial construction labels (only for Chinese) to control the output of the decoder and show the results are better than an unconditioned generation with GPT-2 in terms of several metrics.

Cons and questions:
- The paper describes a generic conditioned language model approach, I can see no novelty of methods or results here.
- No definition of "structures" given. Instead, the authors use fuzzy formulations like "multiple types of multi-granularity structure information" or "the auxiliary model can be any credible model or tool that can extract soundable structure information from the template." In fact, the paper provides the results of experiments with POS labels and participial construction labels (only for Chinese).
- No comparisons with other known syntax-aware language models are done (consider [arXiv:1909.02273], DOI:10.18653/v1/N19-1118, DOI:10.1109/IALP48816.2019.9037672, etc).
- No exact details on the structure encoder or decoder pre-training or finetuning process are given, although they can represent some practical interest.
- No clear description of the resulting architecture is provided ("We only modified the input representation and few parameters of transformer").
- The reason for using a Transformer encoder is ambiguous ("both the transformer encoder and decoder are adopted here is that we want each token in sequence to aware its local and global structure information"), the logic of the number of Encoder layers selection is hard to check without details of the training process.
- The Structure Matching evaluation was done through human assessment. However, no clear rules of the process are given.

Minor comments:
The paper has a lot of typos and grammar issues. Here are some samples:
(1) "importan-t" -- an incorrect word-wrap.
(2) "quality ( like Fluency" -- a space after the open parenthesis
(3) "extracted structure information are regarded"
(4) "word can aware the global structures"
(5) "which can be any precede words for continue generation"
(6)	"reagrded"

There are many more, I suggest at least to use some automatic spellchecking tools.

---

### Official Review · AnonReviewer3 · 2020-10-29
**The paper proposed a text generation method which can utilize the language structures from character-level, word-level to sentence-level structure.**

**Rating:** 5
**Confidence:** 4

**Review:**

The paper proposed a text generation method which can utilize the language structures from character-level, word-level to sentence-level structure.  The proposed model, structure-aware transformer (SAT), explicitly incorporates multiple types of multi-granularity structure information to guide the text generation with corresponding structure.

Pros:
1. The paper is clearly written. The experiments show the effectiveness of the proposed method.
2. The  proposed method can explicitly incorporate multiple types of multi-granularity structure information to guide the text generation.
3. The proposed method shows its advantages in structure control.

Cons:

1. The method just incorporates structure information in the encoder part rather than the decoder. It's hard to guarantee the quality of structure control. Moreover, this way to incorporate structure information is just adding extra features and not a new framework.
2. The structure information used in the paper is only segmentation and POS information. Other information, such as syntax, also should be considered.  Segmentation and POS information are just shallow structure information.
3. BLEU score for POS or BMES is not suitable. Since the structure information is given, the generated text should be evaluated with other measures, such as F1 score.
4. The fluency could be due to the pre-trained language model GPT2. An experiment should be performed without PLMs.
5. Some related references are missing.

Questions:
1. The title of paper is "Structure Controllable Text Generation", but the proposed method is just to infuse structure information as features. Therefore, the proposed method is more like "structure-infused" rather than "structure controllable".

Missing References:
1. Zhang X, Yang Y, Yuan S, et al. Syntax-infused variational autoencoder for text generation[J]. arXiv preprint arXiv:1906.02181, 2019.
2. Casas N, Fonollosa J A R, Costa-jussà M R. Syntax-driven Iterative Expansion Language Models for Controllable Text Generation[J]. arXiv preprint arXiv:2004.02211, 2020.
3. Wu S, Zhou M, Zhang D. Improved Neural Machine Translation with Source Syntax[C]//IJCAI. 2017: 4179-4185.

---

### Decision · Program_Chairs · 2021-01-07
**Final Decision**

**Decision:**

Reject

**Comment:**

This paper proposes a controllable text generation model conditioned on desired structures, converting a text into structure information such as part of speech (POS) and participial construction (PC). It proposes a “Structure Aware Transformer” (SAT) to generate text and claims better  PPL and BLEU compared with GPT-2. Reviewers pointed out that limited novelty of this paper - SAT is essentially a transformer run on multiple sequences of structure information, with sums of structure embeddings as input embeddings -  the proposed method essentially infuses structure information as features, rather than “controlling” text generation. Some references are also missing, most prominently:

1. Zhang X, Yang Y, Yuan S, et al. Syntax-infused variational autoencoder for text generation[J]. arXiv preprint arXiv:1906.02181, 2019.
2. Casas N, Fonollosa J A R, Costa-jussà M R. Syntax-driven Iterative Expansion Language Models for Controllable Text Generation[J]. arXiv preprint arXiv:2004.02211, 2020.
3. Wu S, Zhou M, Zhang D. Improved Neural Machine Translation with Source Syntax[C]//IJCAI. 2017: 4179-4185.

Unfortunately, no answers are provided by the authors to the questions asked by the reviewers, which makes me recommend rejection.